# Nature-Inspired Nanoparticles as Paclitaxel Targeted Carrier for the Treatment of HER2-Positive Breast Cancer

**DOI:** 10.3390/cancers13112526

**Published:** 2021-05-21

**Authors:** Celia Nieto, Milena A. Vega, Eva Martín del Valle

**Affiliations:** Chemical Engineering Department, Faculty of Chemical Sciences, University of Salamanca, 37008 Salamanca, Spain; mvega@usal.es

**Keywords:** breast cancer, HER2-overexpression, drug delivery system, polydopamine nanoparticles, paclitaxel, trastuzumab

## Abstract

**Simple Summary:**

Great advances have been made in the treatment of an aggressive subtype of breast cancer, known as HER2-positive subtype. However, the antitumor therapy most widely employed in the clinical fight against it has severe side effects, and the apparition of treatment resistances is frequent. To overcome these drawbacks, nano-sized particles had already been developed as targeted vehicles for the drug and the antibody that are usually administered as first-line treatment for this subtype of breast cancer. These nanoparticles showed better results than the drug that they transported. Nevertheless, to further reduce the drug effective dose and its toxicity to normal tissues, other nanoparticles, more advantageous than the previous ones, were developed in the current work. Compared to the previous nanoparticles, those prepared here proved to be more efficacious, and their potential administration may constitute an excellent approach to improve the outcomes of patients suffering from HER2-positive breast cancer.

**Abstract:**

Despite the advances made in the fight against HER2-positive breast cancer, the need for less toxic therapies and strategies that avoid the apparition of resistances is indisputable. For this reason, a targeted nanovehicle for paclitaxel and trastuzumab, used in the first-line treatment of this subtype of breast cancer, had already been developed in a previous study. It yielded good results in vitro but, with the aim of further reducing paclitaxel effective dose and its side effects, a novel drug delivery system was prepared in this work. Thus, polydopamine nanoparticles, which are gaining popularity in cancer nanomedicine, were novelty loaded with paclitaxel and trastuzumab. The effectiveness and selectivity of the nanoparticles obtained were validated in vitro with different HER2-overexpressing tumor and stromal cell lines. These nanoparticles showed more remarkable antitumor activity than the nanosystem previously designed and, in addition, to affect stromal cell viability rate less than the parent drug. Moreover, loaded polydopamine nanoparticles, which notably increased the number of apoptotic HER2-positive breast cancer cells after treatment, also maintained an efficient antineoplastic effect when validated in tumor spheroids. Thereby, these bioinspired nanoparticles charged with both trastuzumab and paclitaxel may represent an excellent approach to improve current HER2-positive breast cancer therapies.

## 1. Introduction

Throughout 2020, more than 2.2 million new cases of breast cancer (BC) were diagnosed worldwide and nearly 0.7 million people died from this complex disease, which was the type of cancer with the highest incidence last year [1]. 

Among the new cases of BC that are diagnosed, around 15–20% overexpress the human epidermal growth factor receptor-2 (HER2) [2]. HER2, also overexpressed in other types of solid tumors [3,4], can dimerize with other members of its receptor family, and this fact leads to cancer cell proliferation and survival, angiogenesis and metastasis [4]. In this way, HER2-overexpression may result in more aggressive clinical behavior and has been linked to worse patient outcomes [2]. However, as the overexpression of this tyrosine kinase (TK) receptor does not occur under normal physiological conditions [3], on the other hand, it has allowed the development of anti-HER2 targeted agents that have dramatically improved the survival rate of patients suffering from HER2-positive (HER2+) BC [2]. Undoubtedly, the most widely used HER2-targeted agent to date has been trastuzumab (Herceptin^®^, Basilea, Switzarland), a humanized monoclonal antibody (mAb) approved by the major drug regulatory agencies for the past two decades for the treatment of both early and metastatic BC [5]. Trastuzumab (Tmab) has been shown to induce tumor regression through different molecular mechanisms and to significantly increase overall survival of patients in multiple trials [6,7]. For this reason, it is nowadays an established part of adjuvant HER2+ BC treatment along with taxanes [6,8]. Nonetheless, Tmab intravenous administration also has some downsides. The two most important ones are its inherent cardiotoxicity and the apparition of acquired Tmab resistances, which cause 15% of patients to relapse after treatment [7,9]. 

Likewise, these two drawbacks (severe side effects and apparition of drug resistances) are also characteristic of conventional chemotherapy agents, mainly due to their poor specificity towards neoplastic tissues [10,11]. For instance, paclitaxel (PTX), which is frequently administered as adjuvant therapy with Tmab for the treatment of HER2+ BC [12], is very effective in stabilizing the microtubules of cancer cells and triggering subsequent apoptosis. Nevertheless, it has very poor aqueous solubility and low bioavailability, and the PTX formulation that is normally administered in the clinical setting (Taxol^®^, NY, USA) has severe adverse effects [13,14]. Thereby, the targeted delivery of this antimitotic compound is a major necessity [11,14], and it constitutes one of the main goals that nanotechnology is trying to achieve by means of the development of novel drug delivery systems (DDS) [13,14,15].

With regard to the latter and given that the combination of PTX plus Tmab is regulated and widely employed in the clinical setting, a DDS composed of alginate and piperazine nanoparticles (NPs) was already developed as vehicle for both the mAb and the taxane in a previous work to target HER2+ cancer cells. To enhance PTX aqueous solubility, this drug was previously included in β-cyclodextrin molecules. The resulting complexes along with Tmab were attached to the mentioned NPs, which were not toxic to human tumor or healthy cells. The nanosystem developed showed greater effectiveness and selectivity than the parent drug [16]. Nonetheless, with the aim of further improving the antitumor activity of the taxane, reducing its effective dose and lowering its adverse effects in an even more remarkable manner, a new DDS for Tmab and PTX was developed in the current work.

This DDS is composed of polydopamine (PDA) NPs, which have acquired great relevance in cancer nanomedicine in recent years thanks to their excellent physicochemical properties [17,18]. More specifically, PDA is biocompatible and biodegradable, its chemical versatility permits further functionalization strategies, and it has a strong ability to adsorb drugs via π-π stacking or hydrogen bond interactions. In addition, PDA NPs are capable of releasing drugs in a pH-responsive manner and [19,20,21], most importantly, they have proven to have intrinsic antitumor activity [17,22,23].

So far, no studies were found in the literature in which PTX has been loaded to PDA NPs, and since these NPs are capable of charging drugs with high loading rates, this fact was used to directly load PTX to them. Thus, the employment of additional compounds necessary to improve the taxane solubility, like β-cyclodextrins was avoided by using these NPs, which made the conjugation process more profitable. Furthermore, since PDA NPs have antitumor activity themselves, especially when they are synthesized with 2-propanol [23], synergy between them and PTX was expected. 

Otherwise, Tmab was also directly adsorbed in PDA NPs for the first time. In addition, since PDA can be easily functionalized, the mAb was bound by means of the carbodiimide chemistry to try to enhance the receptor-mediated endocytosis and selectivity of the resulting loaded PDA NPs. 

Thereby, two different nanosystems were obtained, whose antitumor activity and neoplastic selectivity were validated by conventional viability assays. These assays were performed with two human HER2+ BC cell lines and with stromal cells. Based on the results obtained, since PDA NPs covalently charged with Tmab had slightly better antitumor effect and reduced the viability of normal cells less, they were chosen to perform further studies. Herein, their cytotoxicity was analyzed by alive/death confocal microscopy (CLSM) assays and, to verify that PTX pharmacological activity did not change upon loading to PDA NPs, NP ability to induce apoptosis in HER2+ cancer cells was verified by flow cytometry. Furthermore, the effectiveness of the loaded PDA NPs was also validated in HER2+ tumor spheroids to prove that it was maintained in these 3D biostructures. For this purpose, the interaction between the spheroids and the loaded PDA NPs was studied by scanning electron microscopy (SEM). Moreover, additional alive/death CLSM assays and cell counting experiments were carried out. As a result, it was proven that PDA NPs loaded with both PTX and Tmab, were able to significantly reduce the survival rate of HER2+ malignant cells cultured in both 2D and 3D, to a similar or even greater extent than equivalent concentrations of free PTX. Moreover, loaded PDA NPs were notably less toxic to stromal cells than the parent drug. Therefore, their employment may constitute a good strategy to improve PTX bioavailability, while reducing its effective dose and its adverse effects. Likewise, avoiding the apparition of treatment resistances may be possible with the administration of these targeted PDA NPs, which proved to be more advantageous than the previous, similarly reported paclitaxel and trastuzumab DDS.

## 2. Results

### 2.1. Synthesis and Characterization of PDA NPs

Initially, PDA NPs were prepared by the oxidation and self-polymerization of dopamine in a basic, aqueous medium containing 2-propanol (2-PrOH) and NH_4_OH. Even though ethanol is the alcohol normally used in this synthesis procedure [17,23,24], 2-PrOH was chosen because it had been found in a previous work that its employment also allowed PDA NPs to be obtained, and that it conferred them greater antitumor activity in comparison to other alcohols [23]. Otherwise, the NH_4_OH concentration was adjusted in order to prepare NPs with a diameter smaller than 200 nm, since PDA NPs with that size had been observed to have greater inherent antineoplastic effect than bigger ones [22]. 

Once obtained, PDA NP hydrodynamic diameter was determined by dynamic light scattering (DLS). It was found to be 179.3 ± 41 nm (PDI = 0.07) (Figure 1a). In addition, PDA NP surface charge (zeta potential), which was–27.4 ± 1.5 mV, was also determined, and PDA NPs were characterized by transmission electron microscopy (TEM) to analyze their morphology and to obtain a size–range histogram (Figure 1b). According to this, NPs had an average diameter of 124.3 ± 19.2 nm, which was a lower value than that observed by DLS possibly due to the dehydration suffered by the NPs when preparing TEM samples [25].

### 2.2. PDA NP Loading with PTX and Tmab

Next, synthesized PDA NPs were washed through five centrifugation–redispersion cycles in phosphate buffer saline (PBS) and charged with PTX and Tmab following two different methods (Figure 2). First, the drug and the antibody were loaded to PDA NPs through a simple chemisorption process (PDA NPs@Tmab@PTX) by adding solutions of both compounds to PDA NP suspensions. Secondly, the carbodiimide coupling strategy was used to covalently bind Tmab to PDA NPs with the aim of verifying whether this type of conjugation improved the antineoplastic activity and selectivity of the resulting NPs (PDA NPs•Tmab@PTX). To do this, in order for the carbodiimide chemistry to be more efficient, the pH of the NP suspensions was acidified [26], and carboxyl groups of PDA NP surface were activated with EDC and NHS solutions [5]. Later, in two sequential steps, Tmab and PTX solutions with a similar concentration to those used in the adsorption process were incorporated into NP suspensions. 

Charged PDA NPs were isolated by centrifugation, and Tmab and PTX loading efficiencies were both determined by difference, measuring the absorbance of the supernatants obtained by UV-Vis spectrophotometry. Results were obtained according to the two different Tmab conjugation strategies followed and are summarized in Table 1. As can be seen, there were no very significant differences between PTX loading efficiency values, but Tmab conjugation efficiency was greater when the carbodiimide chemistry was employed.

Moreover, zeta potential of PDA NPs@Tmab@PTX and PDA NPs•Tmab@PTX was also analyzed after the conjugation processes. It can be consulted in the Appendix A. 

### 2.3. In Vitro Evaluation of the Antitumor Activity and Selectivity of PDA NPs Charged with Tmab and PTX

In order to evaluate and compare the antitumor activity and selectivity of the two types of loaded PDA NPs obtained, MTT assays were carried out with two breast carcinoma cell lines overexpressing HER2 (BT474 and SKBR3) and with stromal cells (HS5 cell line) [16]. 

For such assays, all cells were treated with 0.035 and 0.042 mg/mL PDA NP concentrations. As already mentioned, these NPs have intrinsic antineoplastic activity, and it had been proven that these two concentrations were able to reduce tumor cell viability in a remarkable manner in previous studies [23]. Likewise, the same concentrations of PDA NPs charged only with PTX or/and Tmab were administered, and cells were also treated with concentrations of the drug and antibody equivalent to those loaded to PDA NPs. To perform the viability assays, a previously developed protocol that avoids overestimating cell viability by subtracting PDA contribution to the absorbance values of the samples was followed [17]. Results obtained after cell treatment with PDA NPs in which Tmab was adsorbed can be found in Appendix A, while those obtained after the administration of PDA NPs covalently conjugated with Tmab are summarized in Figure 3.

As can be seen in both figures, bare PDA NPs were not excessively toxic to normal cells, but they notably reduced BC cell survival rate. These results were in line with those obtained in prior research, in which it had been hypothesized that the great affinity of PDA NPs for the iron (III) existing in the lysosomes could be responsible for their cytotoxicity, which could be thus potentially related to a ferroptosis-mediated excessive production of reactive oxygen species (ROS) in tumor cells [22,23,24].

Otherwise, both PDA NPs●Tmab@PTX and PDA NPs@Tmab@PTX were found to be highly effective, since they reduced the viability of HER2+ BC cells to a slightly greater extent than free PTX concentrations equivalent to those adsorbed in most of the conditions studied (*p* < 0.05) (Appendix A). In this manner, when BT474 cells were treated with PDA NPs•Tmab@PTX, only 10–17% of them survived after 72 h, while this percentage was 13–19% when PDA NPs@Tmab@PTX were employed. Regarding the viability of the SKBR3 cell line, it was 10–14% 72 h after treatment with PDA NPs•Tmab@PTX while, when PDA NPs@Tmab@PTX were administered, SKBR3 survival rate was close to 15–23%. Furthermore, it was noticed in both cases that Tmab covalent conjugation or adsorption to PDA NPs enhanced their antineoplastic activity, since it was greater when NPs were simultaneously charged with the antibody and the drug than when they were only loaded with PTX. Thus, PDA NPs•Tmab@PTX and PDA NPs@Tmab@PTX reduced BT474 and SKBR3 cell viability rates between 5–15% more than PDA NPs@PTX (depending on the NP concentration and treatment time), a fact that revealed that Tmab loading could enhance NP endocytosis in HER2+ tumor cells. 

On the contrary, Tmab loading to PDA NPs made them less toxic to stromal cells than PDA NPs charged only with PTX. Thereby, when HS5 cells were treated with PDA NPs•Tmab@PTX and PDA NPs@Tmab@PTX, their survival rate was 23–30% higher than when the different concentrations of PDA NPs@PTX were used. With respect to the free parent drug, this gap was 25–34% and, compared to BC cells, about 26–36% more stromal cells survived after the same treatment. In fact, Tmab loading even slightly improved the intrinsic toxicity of bare PDA NPs to normal cells.

Even though the differences in terms of efficacy and selectivity between PDA NPs@Tmab@PTX and PDA NPs•Tmab@PTX were not considered statistically significant (*p* > 0.05) (Appendix A), those NPs that had the antibody covalently bound decreased BT474 and SKBR3 viability by 1.2–13.5% more (depending on concentration and time) than PDA NPs@Tmab@PTX. Likewise, PDA NPs•Tmab@PTX affected HS5 survival rate by 2.2–15.9% less than PDA NPs@Tmab@PTX. For this reason, PDA NPs•Tmab@PTX were chosen to perform further studies. 

Herein, their antineoplastic activity was also validated by alive/death CLSM assays, which were performed with the BT474 cell line. These HER2-overexpressing cells were treated with 0.035 mg/mL PDA NPs•Tmab@PTX and with a concentration of free PTX equivalent to that adsorbed to compare the results. Different CLSM images were acquired once 48 and 72 h elapsed. These images have been included in Figure 4a and, from them, the average percentage of living and dead cells was determined for each treatment condition. While 48 and 72 h after treatment with free PTX 36.1% and 16.7% of BT474 cells remained alive, 36.3% and 14.3% of them survived when treated with 0.035 mg/mL PDA NPs•Tmab@PTX for 48 and 72 h respectively. Thereby, results of these CLSM assays proved that PDA NPs•Tmab@PTX were highly effective. 

Finally, given that PTX exerts its pharmacological activity by inducing apoptosis of tumor cells [14], the number of early and late apoptotic BT474 cells was determined by flow cytometry 48 h after treatment with PDA NPs•Tmab@PTX (0.035 mg/mL) and free PTX (99.5 nM). The aim was to verify that the taxane was mainly responsible for the antineoplastic effect of the loaded PDA NPs, and that its adsorption and the later Tmab conjugation to PDA NPs had not altered its antitumor activity. A FITC Annexin V apoptosis detection kit with 7AAD was used [27]. It was observed that treatment with PDA NPs•Tmab@PTX increased the number of early apoptotic HER2+ breast tumor cells (7AAD^−^/Annexin V^+^) in a similar manner than treatment with an equivalent concentration of free PTX (Figure 4b). Therefore, PTX pharmacological activity was not significantly reduced upon loading to PDA NPs. 

### 2.4. Assessment of the Antitumor Effect of PDA NPs Loaded with Tmab and PTX in 3D Cell Cultures

Since in vitro 3D culture systems mimic the in vivo growth conditions of solid tumors better than conventional monolayer cells cultures [28], they have been largely adopted as a model for drug and nanomedicine screening [29,30]. Among the numerous spherical cancer models that can be developed, multicellular tumor spheroids (MCTS), which are obtained by growing cancer cell lines under non-adherent conditions, have been reported to accurately mimic the drug sensitivity behavior of in vivo tumors [29,31]. For this reason, in the current work, developing MCTS derived from the BT474 cell line was decided to further evaluate the antitumor efficacy of the PDA NPs•Tmab@PTX. 

Thus, BT474 cells were assembled in a 3D architecture after seeding them in U-shaped-bottom microplates. By day 3 of growth, they had formed compact multicellular spheres about 1 mm in diameter. At that time, they were treated with PDA NPs•Tmab@PTX (0.035 mg/mL) and with free PTX in an equivalent concentration to that adsorbed to PDA NPs. Three different types of studies were performed.

First, with the aim of analyzing the interaction between the loaded PDA NPs and the 3D cell biostructures, SEM was employed to acquire images of spheroid sections in a secondary electron mode after critical point drying [30]. These images were taken 48 and 72 h after treatment. Morphological differences between the control spheroids and the spheroids treated with both free PTX and loaded PDA NPs could be observed, especially when 72 h had elapsed. In this way, after treatment with the taxane, BT474 spheroids became smaller and disaggregated. This reduction in size and the structural disorganization with loss of integrity could already be detected after exposure to PDA NPs•Tmab@PTX, with which spheroids acquired a grape bunch-shape in which individual cells could be better distinguished (Figure 5). Moreover, in both the images obtained from the spheroids treated with free PTX and with PDA NPs•Tmab@PTX, filamentous-like structures were observed that might correspond to PTX crystals formed in solution [32].

Second, additional live/death CLSM assays were also carried out with the 3D cell cultures (Figure 6a). Again, confocal images were taken 48 and 72 h after spheroid treatment, just as with the monolayer cell cultures. Control images allowed an appreciation of how BT474 MCTS had a necrotic core surrounded by a viable rim, with an inner layer of quiescent cells and an outer layer of proliferating cells. This phenomenon occurs in large tumor spheroids (>500 µm) [29]. Otherwise, the results obtained by SEM were corroborated with these assays, since it could be noticed that both free PTX and PDA NPs•Tmab@PTX, were able to reduce the size of the spheroids developed and to disaggregate them, especially after 72 h. 

Likewise, the effect of PDA NPs•Tmab@PTX treatment on spheroid morphology was borne out by phase-contrast images, which can be found in the Appendix A. From them, it could be observed how after 48 h, loaded PDA NPs made the spheroids smaller and decreased their density. In addition, their margins were less marked, an indication of the cellular disassemble that was beginning to take place. 

Lastly, the viability of the BT474 cells that made up the spheroids was determined by cell-counting 48 h after treating them with free PTX and PDA NPs•Tmab@PTX, as well as with 0.035 mg/mL bare PDA NPs. Results obtained can be found in Figure 6b. As can be seen, the viability of the control spheroids was close to 71 ± 4% due to the existence of the necrotic core described in the previous paragraph, caused by the diffusion gradient of oxygen and nutrients [29]. The mean survival rate of the spheroids treated with unloaded PDA NPs was about 65 ± 5% while, when free PTX and PDA NPs•Tmab@PTX were administered, their viability was reduced to 58 ± 5% and 59 ± 6%, respectively. Therefore, it was proven that loaded PDA NPs were not only effective in inducing apoptosis of monolayer HER2+ BC cells. When these cells were forming 3D structures, they also showed a cytotoxicity similar to that of the parent drug, even spreading worse [33]. 

## 3. Discussion

In the last couple of decades, nanotechnology, taking advantage of the deeper knowledge that the scientific community has acquired about cancer biology, has become essential for the development of DDS that seek to overcome the drawbacks of conventional chemotherapy [10,11,15]. Thus, for instance, with the aim of reducing Tmab and PTX side effects while potentially avoiding the apparition of resistances to these agents, we had already designed alginate and piperazine NPs as a targeted nanovehicle for the anti-HER2 mAb and the taxane [16]. In a similar manner, other authors developed targeted micelles to transport Tmab and PTX to improve HER2+ BC treatment [33]. In both cases, the polymeric NPs and the micelles were inactive, so they did not show cytotoxicity. Moreover, including PTX into β-cyclodextrins was necessary before loading it to alginate-piperazine nanoparticles, and encapsuling the taxane into the micelles took a two -day procedure [16,34]. Thus, in order to make the PTX loading process easier and more profitable, and further reduce the effective doses and the side effects of this drug, a novel PTX-Tmab targeted DDS was developed in the current study to improve HER2+ BC treatment. 

This novel nanovehicle was made up of PDA NPs, which have specific antineoplastic activity themselves [17,22,23], can directly adsorb drugs with great loading rates [19], and can release these drugs in a pH-sensitive manner [35,36]. Thanks to these three properties, it was expected that the taxane could be directly loaded to the aforementioned NPs, that synergy could occur between PDA NPs and PTX, and that PDA pH-sensitivity could improve the selectivity of the DDS obtained, enhancing that provided by Tmab binding. 

Herein, PDA NPs (approximately 180 nm in size) were prepared and easily loaded with PTX for the first time. Likewise, the nanosystem was novelty decorated by Tmab, which was both adsorbed and covalently bound by means of the carbodiimide chemistry. Two types of loaded PDA NPs were obtained, PDA NPs•Tmab@PTX and PDA NPs@Tmab@PTX, which were shown to be highly effective in vitro, reducing the viability rate of HER2+ BC cells similarly or further than PTX concentrations equivalent to those adsorbed. In addition, compared to the other Tmab-PTX-transporting nanosystems mentioned previously [16,34], loaded PDA NPs were found to have more remarkable antitumor activity, perhaps because their inherent toxicity was added to PTX pharmacological activity. In this manner, PDA NPs loaded with both Tmab and PTX reduced the viability of HER2+ breast tumor cells to a greater or equal extent than the similar reported nanomedicines [16,34], but transporting a smaller amount of PTX and having been administered in a lower concentration to the cells. In addition, loaded PDA NPs were not only effective when validated in monolayer cell cultures, but they were also capable of disintegrating and reducing the viability survival rate of HER2+ breast tumor spheroids.

Otherwise, apart from being effective, charged PDA NPs, which were proven to induce apoptosis in target HER2+ BC cells, reduced stromal cell viability to a lesser extent than similar PTX concentrations. Thus, stromal cell survival rate was notably less affected than that of breast tumor cells upon treatment with loaded PDA NPs, while equivalent PTX concentrations reduced the viability of both types of cells in a similar way. 

Therefore, PTX loading to PDA NPs also carrying Tmab could constitute a good strategy to improve the bioavailability of the drug in an efficient manner while reducing its adverse effects. Furthermore, their administration may even avoid the apparition of potential resistances by targeting HER2+ tumor cells thanks to Tmab presence, as well as by releasing PTX in a pH-responsive way in neoplastic tissues.

## 4. Materials and Methods

### 4.1. Chemicals

Dopamine hydrochloride, NH_4_OH (28–30%), PBS (0.01 M, pH 7.4), N-hydroxysuccinimide (NHS), N-(3-Dimethylaminopropyl)-N’-ethylcarbodiimide hydrochloride (EDC), PTX (from semisynthetic, >97%), Dulbecco’s modified Eagle’s medium (DMEM), fetal bovine serum (FBS, USA origin), thiazolyl blue tetrazolium bromide (MTT reagent) and dimethyl sulfoxide (DMSO) were all supplied by Sigma–Aldrich (Darmstadt, Germany). Penicillin-streptomycin (5000 U/mL), calcein AM, propidium iodide ReadyProbes^TM^ reagent, Gibco^TM^ Trypsin-EDTA (0.25%) and trypan blue stain (0.4%) were obtained from Thermo Fisher Scientific (Eugene, OR, USA). 2-PrOH and HCl (37%) were purchased from PanReac Química S.L.U. (Castellar del Vallès, Barcelona, Spain). Tmab was gifted from the Instituto de Biología Molecular y Celular del Cáncer (Salamanca, Spain) and the FITC Annexin V apoptosis detection kit with 7AAD was obtained from Immunostep (Salamanca, Spain).

### 4.2. Synthesis and Characterization of PDA NPs

PDA NPs about 180 nm in size were prepared by dopamine oxidation and self-polymerization in an alkaline medium. Briefly, an ammonia aqueous solution (2.5 mL) was mixed with 2-PrOH (40 mL) and H_2_O(d) (90 mL) under magnetic stirring at room temperature for 30 min. Dopamine hydrochloride (0.5 g) was dissolved in H_2_O(d) (10 mL) and added to the above solution [23]. The resulting mixture was left to react for 24 h and PDA NPs obtained were isolated by centrifugation. Four centrifugation-redispersion cycles in H_2_O(d) were performed to eliminate any residue, and PDA NPs were finally resuspended in H_2_O(d) in a 2 mg/mL concentration once washed.

Next, to characterize PDA NPs, their hydrodynamic diameter was determined by DLS on the basis of their intensity–average size distribution, along with their zeta potential (Zetasizer Nano ZS90, Malvern Instruments Inc., Royston, Hertfordshire, UK). For such a purpose, PDA NPs were dispersed in Trizma base solution (pH 10.0) in a concentration lower than 0.01% (WT). Moreover, PDA NP size and morphology were also characterized by TEM (Tecnai Spirit Twin, Fei Company, Hillsboro, OR, USA). PDA NPs were again resuspended in H_2_O(d) in a concentration inferior to 0.01% (WT) to prepare the samples and drops of this dispersion were deposited on copper grids with a collodion membrane. NPs were allowed to dry for 24 h and TEM images of at least 300 different PDA NPs were taken with a 120 kV voltage acceleration. To conclude, these images were analyzed (ImageJ software, NIH, Bethesda, MD, USA) to make a NP size–range histogram.

### 4.3. PDA NP Loading with Tmab and/or PTX

In order to subsequently carry out in vitro studies, PDA NPs were washed through five more centrifugation-redispersion cycles in PBS before being loaded with Tmab and PTX. Next, this loading process was performed following two different methods. On one hand, to adsorb both Tmab and PTX, solutions of the antibody (1.35 µM, 17.5 µL) and the drug (0.52 µM, 20.2 µL, 2:1 H_2_O(d)/DMSO) were incorporated to suspensions of PDA NPs (2 mg/mL, 350 µL). The resulting mixtures were kept under orbital shaking (100 rpm) in dark conditions, and loaded PDA NPs were isolated the next day by centrifugation [37]. 

On the other hand, to covalently attach Tmab and adsorb PTX to PDA NPs, the pH of the NP suspensions (2 mg/mL, 1 mL) was adjusted to 4.7–4.8 by dropping HCl (37%). Then, solutions of EDC (193 mg/mL, 10 µL) and NHS (58 mg/mL, 10 µL) were added. NPs were kept under orbital shaking (100 rpm) for 40 min and, after that time, a Tmab solution (1.35 µM, 50 µL) was incorporated into the suspensions. Mixtures were kept under agitation for 3 additional hours and, finally, a PTX solution (0.52 µM, 58 µL, 2:1 H_2_O(d)/DMSO) was also added. Final mixtures were left shaking overnight in dark conditions, and charged PDA NPs were isolated by centrifugation, too [16]. 

The same procedures were carried out to prepare PDA NPs@Tmab, PDA NPs•Tmab and PDA NPs@PTX but, in these cases, only the solution of the antibody or the drug were incorporated to the NP suspensions. All supernatants were preserved to determine Tmab and PTX loading efficiencies and content in PDA NPs.

### 4.4. Determination of Tmab and PTX Loading Efficiencies and Content

While quantification of PTX in PDA NP supernatants was achieved via spectrophotometry at 227 nm (UV-1800, Shimadzu Corporation, Kioto, Japan), Tmab concentration in them was determined using the Pierce^TM^ BCA Protein Assay Kit (Thermo Fisher Scientific, Eugene, OR, USA) following the instructions detailed by the manufacturer. The amount of Tmab and PTX charged in the NPs was determined by difference, and the different loading efficiencies (%) and the antibody and drug content in the charged PDA NPs were found by applying the following equations [38]:(1)$$Loading\;efficiency\;\left(\%\right)=\frac{Weight\;ofantibody\;or\;drug\;found\;loaded\;}{Weight\;of\;total\;antibody\;or\;drug\;used}\times 100$$
(2)$$Antibody\;or\;drug\;content\;(W/W)=\frac{Weight\;of\;antibody\;or\;drug\;found\;loaded\;}{Weight\;of\;loaded\;PDA\;NPs}$$

### 4.5. Cell Culture

BT474, SKBR3 and HS5 cell lines (ATCC, Wessel, Germany) were cultured as instructed. They were grown in DMEM, supplemented with FBS (10% V/V) and antibiotics (100 U/mL penicillin and 100 mg/mL streptomycin) at 37 °C in a 95:5 air/CO_2_ humidified atmosphere.

### 4.6. MTT Assays Performance

Once grown, BT474, SKBR3 and HS5 cells were used to carry out MTT assays. For this purpose, they were seeded into 24-well plates (12,000 cells/well) and incubated to allow attachment. The next day, culture medium in the wells of some plates was replaced by supplemented DMEM containing PBS (for the control), PDA NPs (0.035 and 0.042 mg/mL), PTX (99.5 and 119.5 nM), Tmab (0.45 and 0.54 nM), PDA NPs•Tmab (0.035 and 0.042 mg/mL), PDA NPs@PTX (0.035 and 0.042 mg/mL) and PDA NPs•Tmab@PTX (0.035 and 0.042 mg/mL). Culture medium in the wells of other plates was otherwise replaced by supplemented DMEM containing PBS, PDA NPs (0.035 and 0.042 mg/mL), PTX (94.9 and 113.6 nM), Tmab (0.33 and 0.42 nM), PDA NPs@Tmab (0.035 and 0.042 mg/mL), PDA NPs@PTX (0.035 and 0.042 mg/mL) and PDA NPs@Tmab@PTX (0.035 and 0.042 mg/mL). In both cases, cells were later incubated for 24, 48 and 72 h and, after those times, a MTT solution (5 mg/mL, 110 µL) was added to each well. Cell incubation was performed for an additional hour and formazan crystals were dissolved by adding DMSO (500 µL/well). The optical density value of each well was determined by applying a procedure that was previously set up to subtract PDA contribution to absorbance [17], which was measured in a microplate reader (Zetasizer Nano ZS90, Malvern Instruments Inc., Royston, UK). All data were obtained in triplicate. 

### 4.7. Live/death Fluorecence Staining of BT474 Cells

In order to perform the first alive/death CLSM assays, BT474 cells were seeded (12,000 cells/mL) in glass-bottom dishes (Ibidi, Gräfelfing, Germany) and incubated for 24 h. Subsequently, they were treated with or without PTX (99.5 nM) or PDA NPs•Tmab@PTX (0.035 mg/mL) and incubated again. After 48 and 72 h, cells were incubated with calcein AM (1 µM) and propidium iodide (1 drop/mL) fluorescent dyes for 15 min. BT474 stained cells were imaged by CLSM (Leica TCS SP5, Leica Microsystems, L’Hospitalet de Llobregat, Barcelona, Spain) with an excitation/detection of 494/517 and 535/617 nm, respectively. The average number of living and dead cells after each type of treatment was determined from at least 5 different CLSM images.

### 4.8. Flow Cytometry Analysis 

Flow cytometry data were obtained by an annexin V apoptosis detection kit with 7AAD according to the manufacturer’s instructions (Immunostep, Salamanca, Spain). Briefly, BT474 cells were cultured in dishes (12,000 cells/mL), treated with or without PTX (99.5 nM) and PDA NPs•Tmab@PTX (0.035 mg/mL) and incubated for 48 h. After this time, cells were collected, centrifuged and resuspended in warm PBS. Then, they were washed by warm PBS twice and resuspended in annexin binding buffer (1×) at a concentration of 10^6^ cells/mL. Annexin V (5 µL) and 7AAD (5 µL) were added to the cell samples (10^5^ cells/mL, 100 µL), and these were incubated at room temperature and in dark conditions for 15 min. Stained cells were analyzed by flow cytometry (FACSAria^TM^ III cytometer, BD Biosciences, San José, CA, USA), using FITC (green) for annexin V and PerCP (red) for 7AAD.

### 4.9. Development of BT474 Spheroids and SEM Characterization 

BT474 MCTS were obtained by seeding the mentioned cells in Nunclon^TM^ Sphera^TM^ 96-well, U-shaped bottom microplates (Thermo Fisher Scientific, Eugene, OR, USA) (10,000 cells/mL) and allowing them to grow for 3 days. Then, they were treated with or without PTX (99.5 nM) or PDA NPs•Tmab@PTX (0.035 mg/mL), and 48 and 72 h later spheroids were characterized by SEM (JSM-IT500, Jeol, Tokyo, Japan). To prepare the samples, MCTS were carefully collected and sectioned, deposited on filter paper with poly-L-lysine and fixed overnight at 8 °C in PBS with glutaraldehyde (25%). The next day, they were washed by PBS three times and after that, PBS with osmium (1%) was added to the samples. One hour later, spheroids were washed by H_2_O(d) three times and were progressively dehydrated by a gradient concentration of acetone series. Critical point drying was achieved with CO_2_ and samples were observed by SEM after gold coating. 

### 4.10. Live/Death Fluorecence Staining and Cell-Counting of BT474 Spheroids

Live/death CLSM assays with BT474 spheroids were performed in the same way as those with the monolayer cell cultures, simply developing them on the microplates mentioned in the previous point and depositing them once grown in glass-bottom dishes. 

Otherwise, to assess BT474 spheroid viability by cell-counting, these grown 3D cellular spheres were kept on the U-shaped bottom microplates, where they were treated with or without PTX (99.5 nM), bare PDA NPs (0.035 mg/mL) or PDA NPs•Tmab@PTX (0.035 mg/mL). After 48 h, MCTS were washed by warm PBS, disaggregated with trypsin, collected in DMEM and centrifuged. Cells were resuspended in supplemented DMEM and their viability was determined (InvitrogenCountess II FL Automated Cell Counter, ThermoFisher Scientific) after mixing them with trypan blue. Data were again obtained in triplicate.

### 4.11. Statistical Analysis

Data regarding the size (DLS) and zeta potential values of PDA NPs were the average ± SEM of three different measurements. Data concerning MTT assays were analyzed using an unpaired two-tailed Student *t*-test. *p*-values less than 0.05 were considered to be statistically significant. Displayed viability results were the average ± SEM of three replicates per treatment condition.

## 5. Conclusions 

In conclusion, this report has presented the synthesis of PDA NPs that could act as targeted nanocarriers for PTX and Tmab for the treatment of HER2+ breast tumors. The combination of this taxane and this anti-HER2 antibody is extensively used in the clinical setting as adjuvant therapy and, given its great results, several DDS transporting PTX and Tmab have already been developed. However, neither the taxane or the anti-HER2 antibody have been loaded to PDA NPs to date, and the resulting drug delivery NPs have been shown to reduce normal cell viability rate to a lesser extent than the parent drug and to have a very remarkable antitumor activity in vitro, not only in HER2+ conventional cell cultures, but also in breast tumor spheroids. 

Thus, if Tmab-PTX loaded PDA NPs would also yield good results in vivo in the future, they may constitute an excellent approach to overcome the many disadvantages that the current adjuvant treatment of Tmab and PTX entails, mainly related to their toxicity and the apparition of resistances.

## Figures and Tables

**Figure 1 cancers-13-02526-f001:**
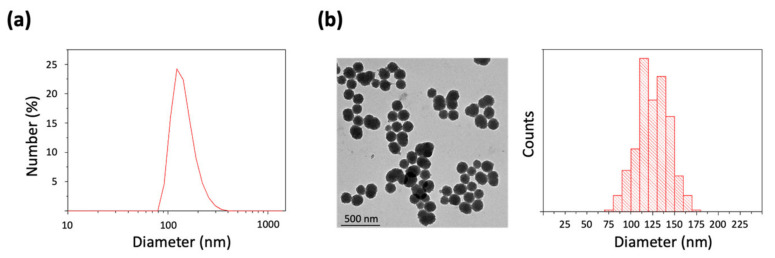
(**a**) DLS number distribution of the PDA NPs synthesized, suspended in Trizma base solution (pH 10.0); (**b**) TEM image and size–range histogram of the PDA NPs obtained.

**Figure 2 cancers-13-02526-f002:**
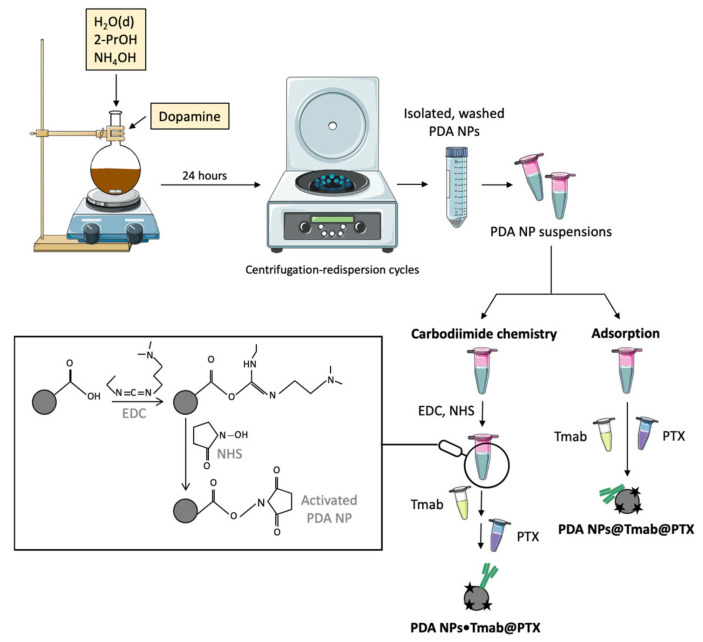
Schematic representation of PDA NPs synthesis and loading with PTX and Tmab by following two different methods in the latter case: adsorption and the carbodiimide coupling chemistry (H_2_O(d)–Deionized water; EDC-N-(3-Dimethylaminopropyl)-N’-ethylcarbodiimide hydrochloride; NHS-N-hydroxysuccinimide), PDA-Polydopamine.

**Figure 3 cancers-13-02526-f003:**
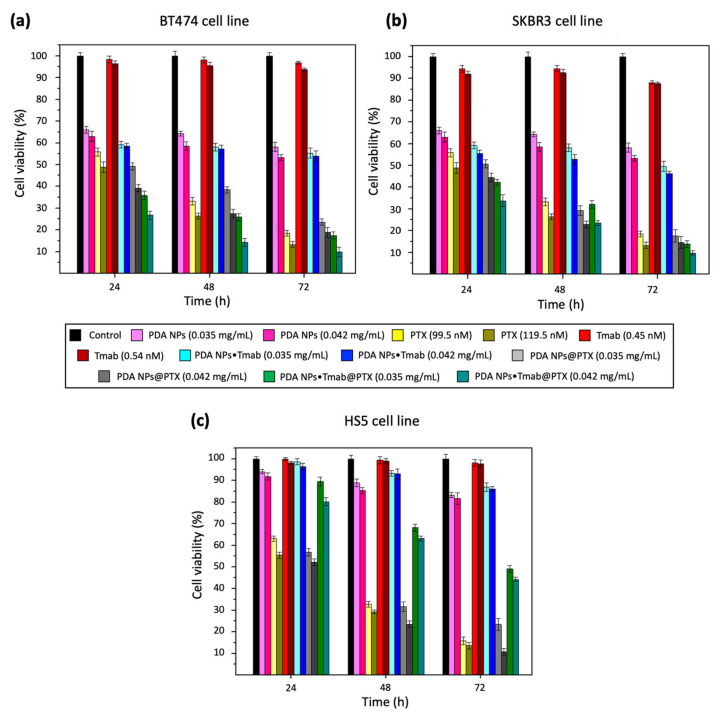
Survival rates of BT474 (**a**), SKBR3 (**b**) and HS5 (**c**) cells after treatment with 0.035 and 0.042 mg/mL bare PDA NPs, PDA NPs•Tmab, PDA NPs@PTX and PDA NPs•Tmab@PTX, as well as with concentrations of free Tmab and PTX similar to those loaded to PDA NPs.

**Figure 4 cancers-13-02526-f004:**
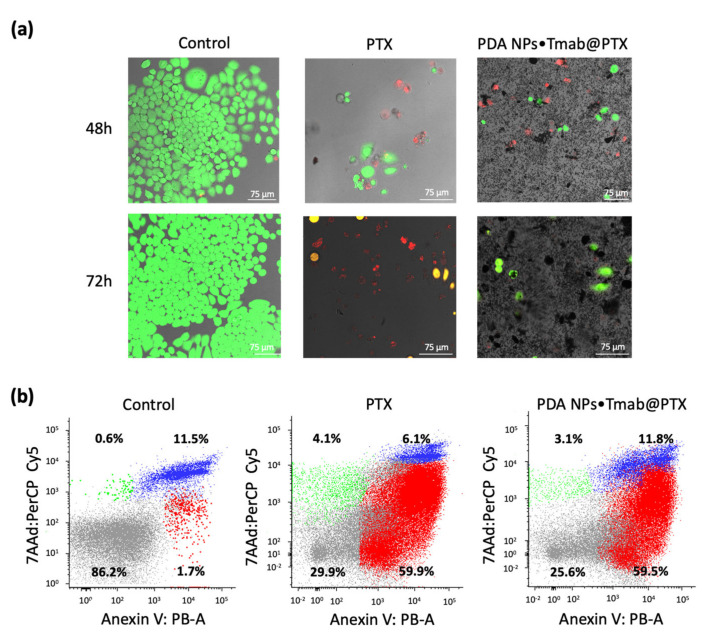
(**a**) CLSM images of BT474 cells 48 and 72 h after treatment with PTX (99.5 nM) and PDA NPs•Tmab@PTX (0.035 mg/mL). Cell viability and death were assessed by using calcein AM (green) and propidium iodide (red), respectively; (**b**) flow cytometry analysis showing the percentage of apoptotic and necrotic BT474 cells 48 h after treatment with PTX (99.5 nM) and PDA NPs•Tmab@PTX (0.035 mg/mL). The vertical axis indicates the cells labeled with 7AAD (PerCP) and the horizontal axis indicates those stained by Annexin V (FITC) (early apoptosis: right lower, late apoptosis/necrosis: right upper).

**Figure 5 cancers-13-02526-f005:**
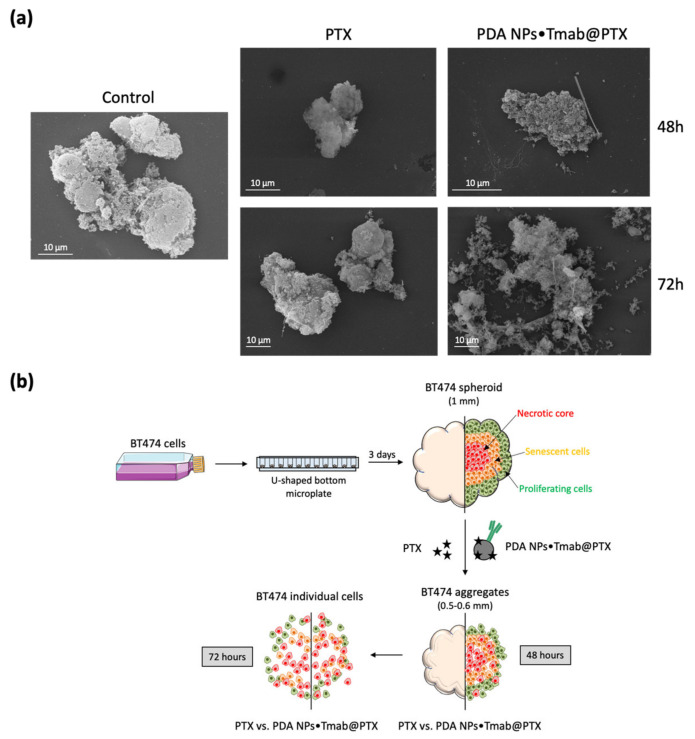
(**a**) SEM characterization of sections of BT474 spheroids 48 and 72 h after treatment with PTX (99.5 nM) and PDA NPs•Tmab@PTX (0.035 mg/mL); (**b**) schematic representation of how BT747 spheroids were developed and the effect that PTX vs. PDA NPs•Tmab@PTX treatment had on their structure and morphology.

**Figure 6 cancers-13-02526-f006:**
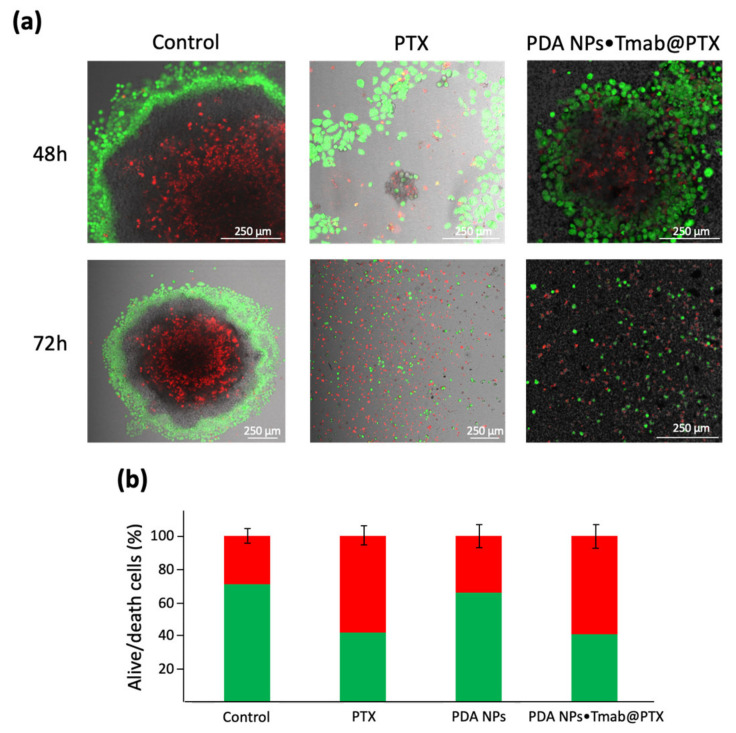
(**a**) CLSM images of BT474 MCTS 48 and 72 h after treatment with PTX (99.5 nM) and PDA NPs•Tmab@PTX (0.035 mg/mL). Cell viability and death were again assessed by using calcein AM (green) and propidium iodide (red); (**b**) percentage of living (green) and dead (red) BT474 cells cultured in 3D spheres, determined by cell-counting, 48 h after treatment with PTX (99.5 nM), PDA NPs (0.035 mg/mL) and PDA NPs•Tmab@PTX (0.035 mg/mL).

**Table 1 cancers-13-02526-t001:** Conjugation efficiencies (%) and Tmab (Trastuzumab) and PTX (Paclitaxel) content in PDA NPs (µg/mg) after adsorbing the drug and loading the antibody by adsorption or by means of the carbodiimide chemistry.

	Adsorption		Carbodiimide Chemistry	
	Conjugation efficiency	Drug/Antibody content	Conjugation efficiency	Drug/Antibody content
**Tmab**	27.4%	1.4 µg/mg NPs	37.8%	1.9 µg/mg NPs
**PTX**	18.0%	2.3 µg/mg NPs	19.1%	2.44 µg/mg NPs

## Data Availability

The data that support the findings of this study are available from the corresponding authors upon reasonable request.

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
