# Peer review of "Nature-Inspired Nanoparticles as Paclitaxel Targeted Carrier for the Treatment of HER2-Positive Breast Cancer"

_cancers, 2021, doi:10.3390/cancers13112526_

Round 1

Reviewer 1 Report

The authors present the results on the synthesis of PDA NPs that could act as nanocarriers for PTX and trastuzumab to treat HER2+ breast tumours. Characterization of PDA NPs was conducted in a very systematic way, including all standard methodologies, i.e. synthesis and characterization of PDA nanoparticles alone, their loading with trastuzumab and paclitaxel, and their effectiveness in cell and spheroids cultures.

1) The authors claim that PDA NPs can be used as selective nanocarriers, but they are not showing the results supporting this. As a reference, normal or non-malignant breast cancer cell line should be treated analogously, and results should be compared.

The experiments with spheroids seem to be unfinished; thus, this part raised some questions:

1) SEM images of spheroids are a bit unusual as they disassemble. This could be the effect of SEM-related sample preparation, but it could also represent cell aggregates, not spheroids. I would expect histological examinations of the spheroids. Images of spheroids in culture should be shown - just phase-contrast image. 

2) Paclitaxel leads to apoptosis; thus, many apoptotic cells in cell culture are reasonable; however, why spheroids after 72h treated with PTX show more alive cells than after 48h. Moreover, Fig.6 show that PDA NPs Tmab@PTX are less efficient than PTX. To compare the effect, I would choose images with the same magnification. Comparing images with a scale bar of 250 um with 75 um is a bit confusing.

3) Staining alive/dead cells typically correlated with fluorescent images, which is not clearly visible in Fig. 6.

4) Only one type of spheroids are shown. How loading PDA Nps with PTX and Tmab affects spheroids created from the other cell lines? 

5) There are some typos in the text, like, e.g. coud instead of could (conclusions).

Author Response

The authors present the results on the synthesis of PDA NPs that could act as nanocarriers for PTX and trastuzumab to treat HER2+ breast tumors. Characterization of PDA NPs was conducted in a very systematic way, including all standard methodologies, i.e. synthesis and characterization of PDA nanoparticles alone, their loading with trastuzumab and paclitaxel, and their effectiveness in cell and spheroids cultures.

Thank you so much for your comments and suggestions. All the changes we have made to the revised version of the manuscript have been marked in red.

1) The authors claim that PDA NPs can be used as selective nanocarriers, but they are not showing the results supporting this. As a reference, normal or non-malignant breast cancer cells should be treated analogously, and results should be compared.

When studying the selectivity of the bare and loaded PDA NPs, it was decided to do it with HS5 cells, as had already been done in previous works [1-3], because fibroblasts are usually reported in the literature as negative control to study the selectivity of nanoparticulate systems. For instance, Bisht et al., Syed et al., and Gao et al. employed NIH/3T3 cells (mouse fibroblasts) to study the tumour-selectivity of ZnO-Fe3O4, silver and selenium nanoparticles, respectively, whose antineoplastic activity was validated with the MDA-MB-231 breast cancer cell line [4-6]. Likewise, Tomankova et al. also used NIH/3T3 cells to assess the specificity of DOX-loaded Fe3O4 nano-assemblies that they validated with MCF7 tumour cells [7], and Tsai et al. employed the mentioned murine fibroblasts to analyse the side toxicity of mesoporous silica nanoparticles with which they treated BT474 cells [8].

In this way, like all the cited authors, we chose stromal cells to assess the selectivity of the PDA NPs loaded with PTX and Tmab. The difference is that, in our case, we chose a stromal cell line of human origin, as we considered that it was better to compare the results obtained with the BT474 and SKBR3 cell lines, also human. Although we believe that the results obtained with the HS5 cell line are enough to prove the specificity of PDA NPs•Tmab@PTX and PDA NPs@Tmab@PTX, if you consider it necessary, we will try to acquire a normal breast cell line to perform new MTT assays whether we are granted 1 or 1.5 months for this.

[1] Nieto C, Centa A, Rodríguez-Rodríguez JA, Pandiella A, Martín del Valle EM. Nanomaterials 2019, 9:948.

[2] Nieto C, Vega M, Enrique J, Marcelo G, Martín del Valle EM. Cancers 2019, 11:1679.

[3] Nieto C, Marcelo G, Vega M, Martín del Valle EM. Colloid. Surf. B Biointerfaces 2021, 199:111506.

[4] Bisht G, Rayamajhi S, KCB, Paudel SN, Karna D, Shrestha BG. Nanoscale Res. Lett. 2016, 11:537.

[5] Syed A, Saraswati S, Kundu GC, Ahmad A. Spectrochim. Acta A Mol. Biomol. Spectrosc. 2013, 114:144-47.

[6] Gao X, Li X, Mu J, Ho C-T, Su J, Zhang Y, Lin X, Chen Z, Li B, Xie Y. Int. J. Biol. Macromol. 2020, 152:605-15.

[7] Tomankova K, Polakova K, Pizova K, Binder S, Havrdova M, Kolarova M, Kriegova E, Zapletalova J, Malina L, Horakova J et al. Int. J. Nanomedicine 2015, 10:949-61.

[8] Tsai C-P, Chen C-Y, Hung Y, Chang F-H, Mou C-Y. J. Mater. Chem. 2009, 19:5737-43.

The experiments with spheroids seem to be unfinished; thus, this part raised some questions:

2) SEM images of spheroids are a bit unusual as they disassemble. This could be the effect of SEM-related sample preparation, but it could also represent cell aggregates, not spheroids. I would expect histological examinations of the spheroids. Images of spheroids in culture should be shown (just phase-contrast images). 

On one hand, the cloud-like morphology of the BT474 spheroids developed, both in the case of the control and the treated spheroids, can also be appreciated in other works in which spheroid SEM images with a scale similar to ours (5-20 µm) were acquired [1,2]. On the other hand, in the images of the spheroids that were treated with both free PTX and PDA NPs•Tmab@PTX, filament-like structures could be distinguished. These “filaments”, based on what has been found in the literature, could correspond to PTX crystals formed in solution [3], and therefore they were not observed in the control. Possibly, by no commenting this fact in the manuscript, it may have seemed that disaggregation of treated spheroids was atypical. Thereby, we have improved the explanation concerning the results obtained when spheroid morphology was analyzed by SEM (lines 275-282). Moreover, we decided to include in Figure 5(a) the images in which the filament structures were more abundant because we found them more “striking”. However, we have now replaced them by others in which we hope that the effect of free PTX and PDA NPs•Tmab@PTX on spheroid disassemble will be better appreciated.

Otherwise, we have now included in the Supplementary Material (Figure S2) a phase-contrast image of an untreated spheroid and a spheroid treated with PDA NPs•Tmab@PTX for 48 hours. In the latter, the results obtained with the alive/death CLSM assay were corroborated: 48 hours after treatment with the loaded PDA NPs, the size of the BT474 spheroid was reduced, its density decreased, and its contour was much less marked because of the cellular disintegration that begging to take place. This explanation has been already included in the manuscript, too (lines 297-301).

Finally, we have found that histological examination of spheroids is usually carried out to examine their morphology or to distinguish the proliferating, quiescent and necrotic cell populations that make up them [4, 5]. Both, the morphology and the cellular subpopulations can be seen well in the phase-contrast images now included in the Supplementary Material and in those obtained by CLSM, respectively, so we believe that histological analysis would not be essential. Anyway, if you consider that this examination is indispensable and must be included in the manuscript, we will find a way to carry it out whether we are given time, since we do not have the necessary instrumentation (for the embedding, cutting and staining the spheroid sections) at the moment.

[1] Mollo V, Scognamiglio P, Marino A, Ciofani G, Santoro F. Adv. Mater. Technol. 2020, 5:1900687.

[2] Yao H-J, Ru R-J, Wang X-X, Zhang Y, Li R-J, Yu Y, Zhang L, Lu W-L. Biomaterials 2011, 32:3285-302.

[3] Castro JS, Tapia LV, Silveyra RA, Martinez CA, Deymier PA. Negative impact of paclitaxel crystallization in hydrogels and novel approaches for anticancer drug delivery systems. In Current cancer treatment: novel beyond conventional approaches, Ozdemir O Eds., IntechOpen Limited: Rijeka, Croatia, 2011.

[4] Ma H-L, Jiang Q, Han S, Wu Y, Tomshine JC, Wang D, Yaling G, Zou G, Liang X-J. Mol. Imaging 2012, 11(6):487-98.

[5] Nautiyal M, Qasen RJ, Fallon JK, Wolf KK, Liu J, Dixon D, Smith PC, Mosedale M. Toxicol. In Vitro 2021, 70:105010.

3) Paclitaxel leads to apoptosis; thus, many apoptotic cells in cell culture are reasonable; however, why spheroids after 72h treated with PTX showed more alive cells than after 48h? Moreover, Fig. 6 shows that PDA NPsTmab@PTX are less efficient than PTX. To compare the effect, I would choose images with the same magnification. Comparing images with a scale bar of 250 µm with 75 µm is a bit confusing.

We are really sorry for the confusion. We would like to thank you for your comment because we made the mistake of including the viability of BT474 spheroids 72 hours after treatment with the PDA NPs•Tmab@PTX and not 48 hours later in Figure 6(b). This error has already been corrected both in the text and in the mentioned figure. Otherwise, possibly the CLSM images we chose in Figure 6(a) were not the most accurate. Following your advice, we have selected others, all of them with the same scale bar, to facilitate the comparison between the control and the two treatments administered (free PTX vs. PDA NPs•Tmab@PTX).

4) Staining alive/dead cells typically correlates with fluorescent images, which is not clearly visible in Fig. 6.

Once the mistake indicated in the previous point has been overcome and other CLSM images have been selected, we think that this correlation is now clearer.

5) Only one type of spheroids is shown. How loading PDA NPs with PTX and Tmab affects spheroids created from the other cell lines? 

The anti-tumour effect of the PDA NPs•Tmab@PTX was only analyzed in BT474 multicellular tumour spheroids (MCTS) because SKBR3 cells are not capable of forming them, they only form loose aggregates [1-3]. Likewise, as our intention was to determine whether PDA NPs•Tmab@PTX maintained the therapeutic activity that they had shown in the MTT assays when administered to 3D cancer cultures, we did no develop MCTS of HS5 cells, despite the fact that these cells are capable of forming spheres [4]. Other authors who have also studied the toxicity of nanoparticulate systems to several cell lines cultured in 2D have only developed MCTS with one of these lines for further studies [5,6]. However, if you consider it necessary to study how PDA NPs•Tmab@PTX treatment would affect aggregates/spheroids developed from the SKBR3 and HS5 cell lines, we will be happy to grow them and carry out the corresponding assays if given enough time.

[1] Ivascu A, Kubbies M. Int. J. Oncol. 2007, 31:1403-13.

[2] Froehlich K, Haeger J-D, Heger J, Pastuschek J, Photini SM, Yan Y, Lupp A, Pfarrer C, Mrowka R, SchlerBner E et al. J. Mammary Gland Biol. Neoplasia 2016, 21:89-98.

[3] Kunz-Schughart LA, Heyder P, Schoeder J, Knuechel R. Exp. Cell Res. 2001, 266:74-86.

[4] Deynoux M, Sunter N, Ducrocq E, Dakik H, Guibon R, Burlaud-Gaillard J, Brisson L, Rouleux-Bonnin F, le Nail L-R, Hérault O et al. PLoS ONE2020, 15(6):e0225485.

[5] Dhanwal V, Katoch A, Singh A, Chakraborty S, Faheem MM, Kaur G, Nayak D, Singh N, Goswami A, Kaur N. Mater. Sci. Eng. C 2019, 97:467-78.

[6] Monteiro PF, Gulfam M, Monteiro CJ, Travanut A, Abelha TF, Pearce AK, Jerôme C, Grabowska AM, Clarke PA, Jerôme C et al. J. Control. Release 2020, 323:549-64.

6) There are some typos in the text, like, e.g. “coud” instead of “could” (conclusions).

Thank you for your appreciation. This error has been corrected, and the entire manuscript has been revised to amend all typos.

Reviewer 2 Report

The manuscript on targeted nanoparticles HER2-positive breast cancer has been well prepared. The technical content has been described comprehensively. The literature is also sufficient and supporting the main content. The potential impact of the Her2 breast cancer is high and may have clinical applications in the future. "accepted" is suggested.

Author Response

The manuscript on targeted nanoparticles for HER2-positive breast cancer treatment has been well prepared. The technical content has been described comprehensively. The literature is also sufficient and supports the main content. The potential impact for the HER2 breast cancer is high and may have clinical applications in the future. "Accepted" is suggested.

Thank you so much for your comments. We really appreciate them.

Reviewer 3 Report

The aim of this study is to determine whether polydopamine nanoparticles loaded with paclitaxel and trastuzumab may represent a great approach to improve therapies for HER2-positive breast cancer. The idea presented by the authors is interesting in a context of cancer therapy. However, the authors should provide more solid evidence to support their interpretations and conclusions.

Major concerns.

Line 169. The authors claim a therapeutic activity of PDA NPs charged with Tmab and PTX. How is the evidence of a therapeutic activity of these compounds on breast cancer disease?

The PDA NPs show toxic effect on breast cancer cells.  However, the molecular mechanism underlying the inhibitory effect of these agents is not deeply investigated so far. In absence of these data the effects of these compounds on breast cancer cells should be better investigated. In addition to the normal stromal cell line HS-5, a normal human mammary epithelial cells should be included in this study in order to better investigate side effects of both PDA NPs and loaded PDA NPs.

Another critical point is that any remarkable differences are observed between the two types of loaded PDA NPs and free PTX, as indicated by data presented in figure 3 and figure S1. A statistical analysis of data must be included. According to data presented in Figure 3 and Figure S1,  what are the advantages of using loaded PDA NPs instead of free PTX?  

The authors wrote “despite the absence of very remarkable differences between PDA NPs@Tmab@PTX and PDA NPs•Tmab@PTX treatments, the latter NPs showed somewhat more pronounced antitumor effect and selectivity”. What data supports this statement? The authors should clarify this point.

Another critical point is that the two types of loaded PDA NPs show toxic effects on HS-5 cells. The data shown in Figure 3 (panel c) and Figure S1 (panel c) and are very impressive. Based on these data, how do you draw the conclusion that these compounds are selective? What is the effect of the two types of loaded PDA NPs in normal cells. Cytostatic or cytotoxic? The authors should deeply discussed this problem. More data are necessary to evaluate the inhibitory effects of both loaded PDA NPs and PDA NPs on normal cells.

Based on data presented in figure 3, the differences between free PDA and PDA NPs•Tmab@PTX should be better investigated by the authors.

Data presented in Figure 4A should be quantified. In addition, the number of early and late apoptotic cells should also be studied in cells treated with PTX free. In addition, based on data reported in Figure 3, which show that the inhibitory effect induced by both PTX free and PDA NPs•Tmab@PTX became stronger with time, this experiment should be also performed after 72h of treatment with PTX free and PDA NPs•Tmab@PTX.

Minor concerns

Line 43: the cited references n 3 and n 4 are not appropriate to support the statement that tyrosine kinase (TK) receptors are overexpressed in other types of solid tumors. Please check and revise.

Line 45: the cited reference n 4 is not appropriate to support the statement that: “…can dimerize with other members of its receptor family, and this fact leads to proliferation, survival, angiogenesis and metastasis of cancer cells”. Indeed, the review (ref n 4) mainly reported on studies of different HER2-targeting strategies and various approaches for HER2-targeted delivery systems to improve outcomes for cancer therapy

Author Response

The aim of this study is to determine whether polydopamine nanoparticles loaded with paclitaxel and trastuzumab may represent a great approach to improve therapies for HER2-positive breast cancer. The idea presented by the authors is interesting in a context of cancer therapy. However, the authors should provide more solid evidence to support their interpretations and conclusions.

Thank you so much for your comments and suggestions. All the changes we have made to the revised version of the manuscript have been marked in red.

Major concerns:

1) Line 169. The authors claim a therapeutic activity of PDA NPs charged with Tmab and PTX. How is the evidence of a therapeutic activity of these compounds on breast cancer disease?

On one hand, although Tmab molecular pathways are not completely known, three main mechanisms have been proposed to be responsible for the antitumor activity of this antibody: (i) inhibition of the PI3K/AKT/mTOR and MEK/ERK signaling cascades; (ii) antibody-dependent cellular cytotoxicity; and (iii) increased production of antiangiogenic factors [1,2]. These three mechanisms are believed to be responsible for the beneficial effects that Tmab has shown to have in the clinical, slowing down tumor progression, inducing tumor regression, and increasing patients’ overall survival rate [2], as indicated in the introduction of the manuscript (lines 53-55).

On the other hand, PTX pharmacological activity has concentration-dependent effects. This taxane is known to stabilize the microtubules of cancer cells, blocking their dynamics and inducing apoptosis, as mentioned also in the manuscript (lines 64-65). At high concentrations, PTX arrests cells in mitosis, and at lower concentrations, it is able to cause a chromosome misseparation on multipolar spindles that leads to tumor cell death [3,4]. Since the FDA authorized PTX administration for breast cancer treatment in 1995, several phase II and III clinical trials have shown that this taxane works well on metastatic breast cancer and has great curative effects [4].

We wonder if you wanted to suggest us with this question that perhaps is not very appropriate to refer to the “therapeutic activity” of the loaded PDA NPs, since results of in vivo assays are not shown in this work. Therefore, along all the manuscript, the expression “therapeutic activity” has been replaced by “antitumor activity” or “antineoplastic activity”.

[1] Valabrega G, Montemuro F, Aglietta M. Ann. Oncol. 2007, 18:977-84.

[2] Nieto C, Vega MA, Martín del Valle EM. Nanomaterials 2020, 10:1674.

[3] Weaver BA. Mol. Biol. Cell 2014, 25:2677-81.

[4] Yang Y-H, Mao J-W, Tan X-L. Chin. J. Nat. Med. 2020, 18(12):890-7.

2) PDA NPs showed toxic effect on breast cancer cells.  However, the molecular mechanism underlying the inhibitory effect of these agents is not deeply investigated so far. In absence of these data, the effects of these nanoparticles on breast cancer cells should be better investigated. In addition to the normal stromal cell line HS5, normal human mammary epithelial cells should be included in this study in order to better investigate the side effects of both PDA NPs and loaded PDA NPs.

In the introduction of the manuscript, it was mentioned that PDA NPs have acquired great relevance in cancer nanomedicine due to their excellent physicochemical properties, and that they have proven to have intrinsic antitumor activity, which may explain their toxicity to breast cancer cells. Both, their physicochemical properties and their cytotoxicity, may be related. Thus, in previous works, it was proven that PDA NPs had great affinity for different metal cations, especially for Ca2+ and Fe3+ [1], and that when they were internalized in breast cancer cells, they remained in their late endosomes/lysosomes [1,2]. Precisely, these cellular organelles are in charge of controlling the intracellular, free Fe2+/3+ concentration by acting as a storehouse, and PDA NPs were believed to be able to chelate the Fe3+ existing in these acidic organelles. In this way, once endocytosed, PDA NPs could cause an imbalance in the intracellular Fe homeostasis, which may trigger an excessive production of reactive oxygen species (ROS) by means of the Fenton chemistry and a subsequent ferroptosis process in cancer cells. This hypothesis was validated by the performance of MTT assays in which BT474 cells were simultaneously treated with PDA NPs and with an Fe chelator compound (DFO) or an antioxidant compound (GSH). Both DFO and GSH counteracted the cytotoxicity of the PDA NPs, so the hypothesis was indirectly proven [2-4]. In addition, in one of the latest published works in which PDA NPs were loaded with Fe3+ to enhance their antitumor activity, it was found that both bare PDA NPs and Fe3+-loaded PDA NPs increased intracellular ROS levels in breast cancer cells that were treated with them [5], further corroborating the described hypothesis.

In the manuscript, we mentioned that the results obtained in the MTT assays concerning the antitumor effect of the bare PDA NPs “were in line with those obtained in prior research, in which it had been hypothesized that PDA NPs may alter lysosomal iron homeostasis of tumor cells once endocytosed”. However, to clarify this point, more information has now been added (lines 189-192).

Otherwise, we only employed the HS5 stromal cell line to analyze the selectivity of the PDA NPs•Tmab@PTX because in most of the studies in which the side toxicity of a nanoparticulate system is investigated, only a stromal cell line is used as negative control [6-10]. In our opinion, the results obtained with the HS5 cell line should be enough to prove the specificity of PDA NPs•Tmab@PTX and PDA NPs@Tmab@PTX, but if you consider it necessary, we will be happy to try to acquire a normal breast cell line to perform new MTT assays whether we are granted 1 or 1.5 months for this.

[1] Vega MA, Nieto C, Marcelo G, Martín del Valle EM. Colloid. Surf. B Biointerfaces 2018, 167:284-90.

[2] Nieto C, Vega MA, Marcelo G, Martín del Valle EM. RSC Adv. 2018, 8:36201.

[3] Nieto C, Vega MA, Enrique J, Marcelo G, Martín del Valle EM. Cancers 2019, 11:1679.

[4] Nieto C, Marcelo G, Vega M, Martín del Valle EM. Colloid. Surf. B Biointerfaces 2021, 199:111506.

[5] Nieto C, Vega MA, Martín del Valle EM. Int. J. Mol. Sci. 2021, 22:3161.

[6] Bisht G, Rayamajhi S, KCB, Paudel SN, Karna D, Shrestha BG. Nanoscale Res. Lett. 2016, 11:537.

[7] Syed A, Saraswati S, Kundu GC, Ahmad A. Spectrochim. Acta A Mol. Biomol. Spectrosc. 2013, 114:144-47.

[8] Gao X, Li X, Mu J, Ho C-T, Su J, Zhang Y, Lin X, Chen Z, Li B, Xie Y. Int. J. Biol. Macromol. 2020, 152:605-15.

[9] Tomankova K, Polakova K, Pizova K, Binder S, Havrdova M, Kolarova M, Kriegova E, Zapletalova J, Malina L, Horakova J et al. Int. J. Nanomedicine 2015, 10:949-61.

[10] Tsai C-P, Chen C-Y, Hung Y, Chang F-H, Mou C-Y. J. Mater. Chem. 2009, 19:5737-43.

3) Another critical point is that any remarkable differences are observed between the two types of loaded PDA NPs and free PTX, as indicated by data presented in Figure 3 and Figure S1. A statistical analysis of data must be included. According to data presented in Figure 3 and Figure S1, what are the advantages of using loaded PDA NPs instead of free PTX?  

Following your directions, we have performed a statistical analysis in order to find out whether the differences between the effect of free PTX and both PDA NPs@Tmab@PTX and PDA NPs•Tmab@PTX on cell viability were significantly different. Thus, we have carried out an unpaired two-tailed Student t-test, considering as a null hypothesis that the mean cell viability values achieved after treatment with the taxane and the two types of loaded PDA NPs was the same and, as an alternative hypothesis, that there were significant differences between the mean viability values obtained after one treatment or another. Thereby, we have rejected the null hypothesis when the t value that we have obtained from our data was higher than the critical value, which in our case was 3.9. The t and p-values obtained for each cell line, treatment condition and time are shown in the following tables. As indicated in the manuscript, we have considered that p-values inferior to 0.05 have been statistically significant.

BT474 cell line

Treatment condition

Time (hours)

t-value

p-value

PTX (99.5 nM) vs. PDA NPs@Tmab@PTX (0.035 mg/mL)

24

6.707

0.0025

48

11.789

0.0003

72

3.112

0.0358

PTX 119.5 nM) vs. PDA NPs@Tmab@PTX (0.042 mg/mL)

24

19.827

3.8171 E-05

48

25.836

1.3332 E-05

72

8.438

0.0011

PTX 119.5 nM) vs. PDA NPs•Tmab@PTX (0.042 mg/mL)

24

12.615

0.0002

48

4.620

0.0098

72

6.241

0.0034

PTX 119.5 nM) vs. PDA NPs•Tmab@PTX (0.042 mg/mL)

24

35.561

3.7323 E-06

48

12.168

0.0003

72

10.515

0.0005

SKBR3 cell line

Treatment condition

Time (hours)

t-value

p-value

PTX (99.5 nM) vs. PDA NPs@Tmab@PTX (0.035 mg/mL)

24

12.028

0.0003

48

10.525

0.0005

72

20.210

3.5388 E-05

PTX 119.5 nM) vs. PDA NPs@Tmab@PTX (0.042 mg/mL)

24

8.038

0.0013

48

8.466

0.0011

72

8.614

0.0010

PTX 119.5 nM) vs. PDA NPs•Tmab@PTX (0.042 mg/mL)

24

8.187

0.0012

48

5.943

0.0040

72

0.412

0.7011

PTX 119.5 nM) vs. PDA NPs•Tmab@PTX (0.042 mg/mL)

24

14.798

0.0001

48

3.280

0.0304

72

3.699

0.0209

HS5 cell line

Treatment condition

Time (hours)

t-value

p-value

PTX (99.5 nM) vs. PDA NPs@Tmab@PTX (0.035 mg/mL)

24

2.145

0.0985

48

9.238

0.0008

72

7.684

0.0015

PTX 119.5 nM) vs. PDA NPs@Tmab@PTX (0.042 mg/mL)

24

0.890

0.4238

48

9.701

0.0006

72

33.161

4.9319 E-06

PTX 119.5 nM) vs. PDA NPs•Tmab@PTX (0.042 mg/mL)

24

5.262

0.0062

48

80.584

1.4213 E-07

72

88.264

9.8774 E-08

PTX 119.5 nM) vs. PDA NPs•Tmab@PTX (0.042 mg/mL)

24

23.727

1.8710 E-05

48

22.825

2.1824 E-05

72

43.853

1.6168 E-06

As can be observed, p-values obtained have been higher than 0.05 only in 3 of the 36 comparisons made, so we believe that there was a significant difference between the effect on cell viability of free PTX treatment and that of the treatment carried out with both types of loaded PDA NPs. Nevertheless, in lines 195-196 of the manuscript, we have emphasized that the mentioned difference was significant (p < 0.05) in most of the conditions studied, thus taking into account the results got in the statistical analysis. If you consider it necessary, we will be glad to include the three tables shown in the Supplementary Material.

Regarding the advantages of using the loaded PDA NPs instead of free PTX, we would like to point out that, in order to systemically administer the taxane in the clinical, it has to be dissolved in a mixture of Cremophor/EtOH that entails high toxicity. However, Tmab-loaded PDA NP employment could improve its targeted transport. Otherwise, charged PDA NPs showed to reduce the viability of breast cancer cells in a similar or slightly more marked manner to free PTX. Nonetheless, they did not affect the survival rate of stromal cells as much as the taxane, so their use could potentially improve the selectivity of HER2+ breast cancer treatment. Furthermore, in vivo, this selectivity may be further increased due to the EPR effect and PDA NP sensitivity to pH, which would contribute to the release of the taxane in the acidic environment that characterizes tumor tissues. Finally, given that, as has been highlighted in other points, PDA NPs may induce ferroptosis in tumor cells, their administration could also hinder the apparition of resistances, which are very frequent when therapies are only based on pro-apoptotic agents. In addition, as loaded PDA NPs are a targeted nanosystem, this targeting would also contribute to reduce the probability of the appearance of treatment resistances.

4) The authors wrote “despite the absence of very remarkable differences between PDA NPs@Tmab@PTX and PDA NPs•Tmab@PTX treatments, the latter NPs showed somewhat more pronounced antitumor effect and selectivity”. What data supports this statement? The authors should clarify this point.

With this statement we wanted to emphasize that, although differences in terms of efficacy and selectivity between PDA NPs@Tmab@PTX and PDA NPs•Tmab@PTX were not very large, those NPs that had the antibody covalently bound decreased BT474 and SKBR3 viability to a slightly greater extent (1.2-13.5% more, depending on concentration and time) than PDA NPs@Tmab@PTX. Likewise, PDA NPs•Tmab@PTX affected HS5 survival rate to a slightly lesser extent (2.2-15.9% less) than PDA NPs@Tmab@PTX. For this reason, PDA NPs•Tmab@PTX were the NPs chosen to perform further studies.

Following your advice, this point has been clarified in the manuscript. Thereby, the percentage differences mentioned here have been included to improve the explanation of why PDA NPs•Tmab@PTX were preferred to carry out the studies that followed the MTT assays (lines 220-226).

5) Another critical point is that the two types of loaded PDA NPs showed toxic effects on HS5 cells. The data shown in Figure 3 (panel c) and Figure S1 (panel c) are very impressive. Based on these data, how do you draw the conclusion that these compounds are selective? What is the effect of the two types of loaded PDA NPs in normal cells, cytostatic or cytotoxic? The authors should deeply discuss this problem. More data are necessary to evaluate the inhibitory effects of both loaded PDA NPs and PDA NPs on normal cells.

Throughout the manuscript, where we pointed out that loaded PDA NPs were selective, we meant that they decreased the viability of stromal cells to a lesser extent than that of breast cancer cells, and that the two types of PDA NPs loaded with Tmab and PTX were more selective than free PTX, whose toxicity to the three cell lines tested (BT474, SKBR3 and HS5) was similar.

When performing the MTT assays with monolayer cell cultures, fibroblasts, like the rest of the cells, were over-nourished and, therefore, an elevated percentage of them also internalized the loaded PDA NPs. Thus, their viability became also reduced. This reduction was expected to be related with the cytotoxic effect of the taxane, since when in previous studies a co-culture of BT474 and HS5 cells was treated with alginate-piperazine NPs conjugated with both Tmab and PTX, HS5 cells were killed, although also to a lesser extent than BT474 cells, as in this case [1].

To be more precise and given that loaded PDA NPs also affect the viability of HS5 cells, in lines 350 and 508 of the manuscript, in which we mentioned that the nanosystem developed was “selective”, we have remarked that loaded PDA NPs were “more selective than equivalent concentrations of free PTX”.

[1] Nieto C, Centa A, Rodríguez-Rodríguez JA, Pandiella A, Martín del Valle EM. Nanomaterials 2019, 9:948.

6) Based on data presented in Figure 3, the differences between bare PDA NPs and PDA NPs•Tmab@PTX should be better investigated by the authors.

According to what has been commented in the issue number 2), we believe that the antitumor action of bare PDA NPs and PDA NPs•Tmab@PTX happened for different reasons. On one hand, as has been mentioned, the antineoplastic effect of the unloaded PDA NPs could be mediated by an imbalance in the intracellular Fe concentration, followed by an excessive production of ROS and a type of regulated cell death known as ferroptosis [1-3]. Herein, ferroptosis is an oxidative, iron-dependent form of cell death that does not occur to the same extent in normal cells because they do not need so much Fe as malignant cells. This type of regulated cell death (RCD), which is triggered by inactivation of the cellular glutathione (GSH)-dependent antioxidant defenses, is distinct from necrosis, autophagy and apoptosis [4,5]. Otherwise, apoptosis is the type of RCD that is induced after PTX treatment. This antimitotic drug is capable of initiating a cascade of signaling pathways resulting in programmed cell death [6], which in turn partially accounted for the cytotoxicity of PDA NPs•Tmab@PTX. The antitumor effect of these loaded NPs could be due, therefore, to the “sum” of the intrinsic cytotoxicity of PDA NPs plus PTX pharmacological action, so that both ferroptosis and apoptosis mechanisms could contribute to PDA NP•Tmab@PTX antitumor effect producing a certain synergy. By contrast, the cytotoxicity of naked PDA NPs, from what we know so far, could be mainly due to a ferroptosis process.

[1] Nieto C, Vega MA, Marcelo G, Martín del Valle EM. RSC Adv. 2018, 8:36201.

[2] Nieto C, Vega MA, Enrique J, Marcelo G, Martín del Valle EM. Cancers 2019, 11:1679.

[3] Nieto C, Marcelo G, Vega M, Martín del Valle EM. Colloid. Surf. B Biointerfaces 2021, 199:111506.

[4] Nieto C, Vega MA, Martín del Valle EM. Int. J. Mol. Sci. 2021, 22:3161.

[5] Cao JY, Dixon SJ. Cell Mol. Life Sci. 2016, 73:2195-209.

[6] Abu Samaan TM, Samec M, Liskova A, Kubatka P, Büsselberg D. Biomolecules 2019, 9:789.

7) Data presented in Figure 4(a) should be quantified. In addition, the number of early and late apoptotic cells should also be studied in cells treated with free PTX. In addition, based on data reported in Figure 3, which showed that the inhibitory effect induced by both free PTX and PDA NPs•Tmab@PTX became stronger with time, this experiment should be also performed after 72h of treatment with free PTX and PDA NPs•Tmab@PTX.

Thank you so much for your suggestions. First, since we acquired between five and eight CLSM images for each treatment condition and measured time reflected in Figure 4(a), we have determined the average number of BT474 living and death cells from them and included the different percentage values obtained in the manuscript (lines 231-236).

Secondly, apoptosis results achieved by flow cytometry when BT474 cells were treated with free PTX for 48 hours have been now included in Figure 4(b), too. Our intention when determining the percentage of early and late apoptotic cells after PDA NPs•Tmab@PTX treatment was only to make sure that the pharmacological activity of paclitaxel was not reduced after being loaded to PDA NPs, and this is why we carried out these experiments at the intermediate time of those studied in the MTT assays. As can be seen now in Figure 4(b), the percentage of early apoptotic cells was very similar after BT474 treatment with free PTX and PDA NPs•Tmab@PTX, so we were able to corroborate that the taxane ability to induce apoptosis had not been reduced after its incorporation to the NPs. However, if you still consider that flow cytometry analysis is necessary after 72 hours, we will perform it without any inconvenience if we are allowed to do so for about an additional month (since the analysis is carried out in an external faculty, and due to the situation triggered by Covid-19, technicians take a long time to have availability).

Minor concerns:

8) Line 43: The cited references n 3 and n 4 are not appropriate to support the statement that tyrosine kinase (TK) receptors are overexpressed in other types of solid tumors. Please check and revise.

The statements “HER2 is overexpressed in a number of human cancers including breast, ovarian, lung, gastric and oral cancers…” and “In human cancers, such as breast, gastric, ovary, prostate, and lung, there is a HER2 gene amplification and overexpression which makes it a promising target for cancer therapy” can be found in references 3 and 4, respectively. Thus, we believe that these two references support the idea that we wanted to convey that HER2 is overexpressed in more types of cancer, not just breast cancer. Perhaps, confusion could have been produced by mentioning “This tyrosine kinase (TK) receptor…” and referring to HER2 generically instead of naming it directly. For this reason, we have replaced these words by “HER2” in the manuscript (line 42).

9) Line 45: The cited reference n 4 is not appropriate to support the statement that: “…can dimerize with other members of its receptor family, and this fact leads to proliferation, survival, angiogenesis and metastasis of cancer cells”. Indeed, the review (ref n 4) mainly reported studies of different HER2-targeting strategies and various approaches for HER2-targeted delivery systems to improve outcomes for cancer therapy.

In the work corresponding to the reference number 4, authors mention in the first paragraph of the introduction section that HERs “can dimerize (forming ten different homo- and heterodimers), which lead to complex biological signaling pathways, such as the phosphatidylinositol 3-kinase/protein kinase B (PI3K/Akt) and mitogen-activated protein kinase/ extracellular-related kinase 1/2 (MAPK/ERK1/2) pathways. These signaling cascades ultimately lead to cell proliferation, survival, migration, angiogenesis (development of new blood vessels), and metastasis. Among these receptors, HER2 (molecular mass of 185 KDa)-mediated heterodimerization is a stable powerful signal transduction pathway among all the dimers formed by HER family”. In this way, we considered that was appropriated to support the statement that we included in the lines 42-44 of the introduction of our work. Nonetheless, if you think that another reference could better reinforce the fact that was described, we will be happy to include it in the manuscript.

Round 2

Reviewer 3 Report

Overall, this second version of the manuscript is improved. The authors addressed almost all the comments and suggestions. However, I only have a few minor comments to make.

The authors claim that the HS5 stromal cell line is used as negative control in most of the studies in which the side toxicity of a nanoparticulate system is investigated. However, to better investigate the side effects of both PDA NPs and loaded PDA a more suitable model of normal human mammary epithelial cells should be taken in consideration. In absence of data, the definition that loaded NPs are selective is not correct. The authors should correct the term “selective”in order to make the statements more comprehensible.

Please include the three tables in the Supplementary Materials

Figure 4: the images should be with the same scale bar. The authors should correct them.

The authors wrote that NPs that had the antibody covalently bound decreased BT474 and SKBR3 viability to a slightly greater extent. This point is still not clear. A statistical significance should be tested to make conclusions.

Author Response

Overall, this second version of the manuscript is improved. The authors addressed almost all the comments and suggestions. However, I only have a few minor comments to make.

Thank you for the comments and suggestions made to improve our manuscript. Again, we have highlighted in red all the changes that we have made when performing the second revision of the manuscript.

1) The authors claim that the HS5 stromal cell line is used as negative control in most of the studies in which the side toxicity of a nanoparticulate system is investigated. However, to better investigate the side effects of both PDA NPs and loaded PDA a more suitable model of normal human mammary epithelial cells should be taken in consideration. In absence of data, the definition that loaded NPs are selective is not correct. The authors should correct the term “selective” in order to make the statements more comprehensible.

According to your comment, the statements in which it was mentioned that loaded PDA NPs were selective have been replaced throughout the manuscript, mentioning instead that nanoparticles decreased the viability of stromal cells to a lesser extent that that of breast cancer cells. Changes can be seen in the lines 17, 28, 103, 114, 356-359 and 515.

2) Please include the three tables in the Supplementary Material

Done. Tables S2-S4 containing the data regarding the statistical analysis performed during the previous revision of the manuscript have been included in the Supplementary Material.

3) Figure 4: the images should be with the same scale bar. The authors should correct them.

Thank you so much for this appreciation. The scale bars of the CLSM images that appear in Figure 4(a) have already been unified.

4) The authors wrote that NPs that had the antibody covalently bound decreased BT474 and SKBR3 viability to a slightly greater extent. This point is still not clear. A statistical significance should be tested to make conclusions.

Following your suggestion, we have performed a new statistical analysis to determine whether the differences between the cytotoxicity of PDA NPs@Tmab@PTX and PDA NPs•Tmab@PTX were statistically significant. We have carried out again an unpaired two-tailed Student t-test, considering as a null hypothesis that the mean cell viability values achieved after treatment with both types of loaded PDA NPs was the same and, as an alternative hypothesis, that there were significant differences between the mean cell viability values obtained after treatment with PDA NPs@Tmab@PTX and PDA NPs•Tmab@PTX. T- and p-values obtained for each cell line, PDA NP concentration and time are shown in the following tables, which have also been included in the Supplementary Material (Tables S5-S7). Anew, the null hypothesis was rejected when the p-values obtained were inferior to 0.05.

BT474 cell line

Treatment condition

Time (hours)

t-value

p-value

PDA NPs@Tmab@PTX vs. PDA NPs•Tmab@PTX (0.035 mg/mL)

24

2.501

0.0667

48

6.065

0.0037

72

4.613

0.0099

PDA NPs@Tmab@PTX vs. PDA NPs•Tmab@PTX (0.042 mg/mL)

24

10.741

0.0004

48

1.475

0.2142

72

2.994

0.0402

SKBR3 cell line

Treatment condition

Time (hours)

t-value

p-value

PDA NPs@Tmab@PTX vs. PDA NPs•Tmab@PTX (0.035 mg/mL)

24

6.313

0.0032

48

2.115

0.1019

72

0.402

0.7080

PDA NPs@Tmab@PTX vs. PDA NPs•Tmab@PTX (0.042 mg/mL)

24

4.926

0.0079

48

0.346

0.7468

72

1.282

0.1425

HS5 cell line

Treatment condition

Time (hours)

t-value

p-value

PDA NPs@Tmab@PTX vs. PDA NPs•Tmab@PTX (0.035 mg/mL)

24

0.088

0.9341

48

3.970

0.0165

72

1.694

0.1654

PDA NPs@Tmab@PTX vs. PDA NPs•Tmab@PTX (0.042 mg/mL)

24

0.395

0.7133

48

9.790

0.0006

72

2.671

0.0557

As shown in the tables above, differences between the cytotoxicity of the PDA NPs@Tmab@PTX and PDA NPs•Tmab@PTX were not statistically significant in a high number of treatment conditions. For this reason, in the previous version of the manuscript, we had emphasized that “Even though the differences in terms of efficacy and selectivity between PDA NPs@Tmab@PTX and PDA NPs•Tmab@PTX were not very pronounced…”. However, as this statement might not be rigorous enough, we have now indicated that such differences “were not statistically significant (p>0.05)” (lines 220-221 of the manuscript).
